# P16-CD8-Ki67 Triple Algorithm for Prediction of CDKN2A Mutations in Patients with Multiple Primary and Familial Melanoma

**DOI:** 10.3390/diagnostics14080813

**Published:** 2024-04-13

**Authors:** Luana-Andreea Nurla, Emma Gheorghe, Mariana Aşchie, Georgeta Camelia Cozaru, Cristian Ionuț Orășanu, Mǎdǎlina Boşoteanu

**Affiliations:** 1Department of Dermatovenerology, “Elias” Emergency University Hospital, 011461 Bucharest, Romania; 2Institute of Doctoral Studies, Doctoral School of Medicine, “Ovidius” University of Constanţa, 900573 Constanta, Romania; 3Department of Dermatology, “Sf. Apostol Andrei” Emergency County Hospital, 900591 Constanta, Romania; 4Department of Histology, Faculty of Medicine, “Ovidius” University of Constanţa, 900527 Constanta, Romania; 5Clinical Service of Pathology, “Sf. Apostol Andrei” Emergency County Hospital, 900591 Constanta, Romania; 6Department of Pathology, Faculty of Medicine, “Ovidius” University of Constanţa, 900527 Constanta, Romania; 7Department VIII—Medical Sciences, Academy of Romanian Scientists, 030167 Bucharest, Romania; 8Center for Research and Development of the Morphological and Genetic Studies of Malignant Pathology (CEDMOG), 900591 Constanta, Romania

**Keywords:** p16, CD8, Ki67, multiple primary melanoma, familial melanoma, CDKN2A

## Abstract

Melanoma, a malignant neuroectodermic tumor originating from the neural crest, presents a growing global public health challenge and is anticipated to become the second most prevalent malignancy in the USA by 2040. The CDKN2A gene, particularly p16INK4a, plays a pivotal role in inhibiting the cell cycle via the cyclin D/CDK2-pRb pathway in certain tumors. In familial melanomas (FM), 40% exhibit CDKN2A mutations affecting p16INK4a, impacting checkpoint G1, and stabilizing p53 expression. This study aims to establish a scoring system using immunohistochemical antibodies, providing a cost-saving approach to classify multiple primary melanomas (MPM) and FM patients based on their mutational status, thus mitigating genetic testing expenses. This retrospective study included 23 patients with MPM and FM, assessing the p16, CD8, and Ki67 immunohistochemical status. Analyses of each parameter and associations between their value intervals and genetic CDKN2A status were conducted. A total score of at least 9 out of 10 points per tumor defined melanomas with homozygous CDKN2A deletions, exhibiting a sensitivity of 100% and specificity of 94.11%. In conclusion, p16, CD8, and Ki67 individually serve as valuable indicators for predicting melanoma evolution. The algorithm, comprising these three immunohistochemical parameters based on their prognostic and evolutionary significance, proves to be a valuable auxiliary diagnostic tool for cost-effective prediction of mutational status in detecting multiple and familial primary melanomas with CDKN2A homozygous deletion.

## 1. Introduction

Melanoma of the skin and mucosa is a malignant neuroectodermic tumor originating from the neural crest, whose significant augmentation in incidence and morbidity placed it among the main global public health problems [1]. Moreover, current statistics indicate the potential of this oncological condition to become the second most frequent malignancy in the USA by 2040 [2].

The alterations of the tumoral suppressive CDKN2A gene (p16^INK4a^) have been demonstrated in the past to play an important role in certain tumoral types [3,4] due to the corresponding p16 protein that acts by inhibiting the cell cycle progression, mediated by its inhibitory effect over the cyclin D/CDK2-pRb pathway [5]. P16 may be inactivated by homozygous deletions [6] or methylation of the promoter region [7], and this phenomenon commonly occurs in families with melanoma aggregation [8], while CDKN2A mutations are registered less frequently in primary tumors associated with sporadic cases [9]. Therefore, 40% of familial melanomas present CDKN2A mutations that determine defects of the p16^INK4a^ protein, with a considerable role in the regulation of checkpoint G1 and the stabilization of p53 expression [10,11]. This protein binds to CDK4 and generates the blockage of the cell cycle; any modification at this level creates disruption in the evolution of the cell cycle [12]. Even though p16^INK4a^ does not constitute a possible therapeutic target for the medication that is currently under development in the field of melanoma, associated proteins, such as CDK4, may be addressed [13].

Recent dermatopathology studies illustrated the major utility of immunohistochemistry in the correct and exhaustive diagnosis of malignant melanocytic tumors, indicating that patients with loss of p16 expression were associated with significantly lower survival rates [14], a finding congruent with those noted in studies exploring other tumoral categories, such as pancreatic carcinoma [15], leukemias [16], and pulmonary carcinoma [17], where p16 alterations are corroborated with superior tumoral aggressivity and unfavorable prognosis. Absent or minimal p16 immunohistochemical staining was significantly associated with the presence of ulceration and vascular invasion in the primary tumors, while recurrence-free survival (RFS) was lower in this category of patients [18]. Moreover, p16 expression and the proliferation rate were suggested as relevant markers of the metastatic potential of melanomas, thus highlighting the potential value of p16 immunophenotype evaluation for the treatment plan of cutaneous malignant melanocytic lesions [19].

The importance of tumoral proliferation examination was emphasized by the quality of Ki67 nuclear agent expression, superior to that of mitotic activity, during the G1, S, and G2 phases of the cell cycle in the cellular proliferative populations [20]. Immunohistochemical techniques use Ki67 to identify its level of expression in relation to cellular proliferative activity, disease progression, and recurrence rates [21]. Certain research studies proved the prognostic value of this parameter in multiple solid tumors, such as non-small cell lung cancer (NSCLC) [22], gastrointestinal stromal tumors [23], glioma [24], and thyroid cancer [25]. Its negative prognostic value in melanoma was indicated, indifferent to other histopathological severity criteria, such as tumoral thickness over 4 mm, vascular invasion, or intense mitotic activity reported per mm^2^ [26,27].

The characterization of the immune tumoral microclimate is currently investigated as a prognostic and predictive biomarker for the response to immunotherapy of solid tumors, including melanoma [28]. The presence of tumor-infiltrating lymphocytes (TILs) was associated with a favorable prognosis in melanoma, irrespective of other clinical and pathological characteristics [29], even though the per se implication of TILs in prognosis improvement remains a controversial concept [30]. However, exploration of the immune tumoral infiltrate facilitates the definition of the melanoma case evolution [31]. 

Given the significant prevalence of melanoma, diagnosis- and treatment-associated expenditure is exponentially augmented; therefore, new patient triage approaches for different types of therapies are necessary. The aim of this study is to define a scoring system based on immunohistochemical antibodies designed to predict cases with various CDKN2A alterations in order to mitigate the costs related to standard genetic testing and classify multiple primary and familial melanoma patients based on their mutational status using cost-effective investigations.

## 2. Materials and Methods

This study focused on a 5-year period (2018–2022) and explored patients admitted to the Clinical Emergency County Hospital “Sf. Apostol Andrei” of Constanta with a confirmed diagnosis of melanoma. Afterward, the digital database and conventional archives were explored in order to select the patients with multiple diagnoses of primary melanomas and those with a history of the aforementioned condition in at least one first-degree relative. The exclusion criteria referred to in situ melanomas, underage patients, those with melanoma detected in visceral sites, without other affected family members, or without previously diagnosed primary melanomas. Twenty-three patients were included in the research, among which 7 were diagnosed with familial melanoma (FM) and 16 with multiple primary melanoma (MPM). Out of the 50 primary tumors that were identified, the most representative one was extracted for each patient, focusing on those with appropriately conserved tissue; the immunohistochemical techniques were applied on each representative tumor of every subject so that the number of examined lesions was equal to that of the patients included in this study. 

Afterward, epidemiological parameters, such as age at diagnosis and sex, clinical characteristics, such as the anatomic site of the primary tumors, and histopathological indicators, such as the subtype of melanoma, Breslow index, and mitotic activity, were noted. 

Immunohistochemical testing comprised p16 (clone MX007, Master Diagnostica, Sevilla, Spain), CD8 (clone SP16, Master Diagnostica, Sevilla, Spain), and Ki67 (clone SP6, Biocare, Pacheco, CA, USA). The intensity of the staining resulting after the application of the p16 marker was classified as absent (−), mild (+), moderate (++), and intense (+++), but this parameter was not taken into consideration for the in-depth study. However, p16 was considered positive when displaying intense nuclear and cytoplasmic reactions, and the percentage of p16-stained cells was counted. The segregation of p16 positivity in the examined cohort was performed similarly to the one proposed by the team conducted by Uguen, as follows: >50% positivity; 11–50% positivity; 1–10% positivity; and complete absence of p16 immunoreaction [32]. The CD8 staining was also performed on 4-µm sections derived from formalin-fixed paraffin-embedded specimens, employing the heat-induced epitope retrieval method for epitope discovery. The evaluation of TILs was made in a specific region of focus chosen in the tumor invasion area, and the percentage of CD8-positive TILs was then reported. The CD8 intensity was classified as weak (5–20%), moderate (20–60%), and intense (>60%), according to the study of Kavvadas et al. [33]. Finally, the Ki67 index was determined in the area with the most intense proliferation and with the absence of major inflammatory infiltrate, criteria that were assessed at low magnification (×10), in “hot spot”, and the percentual results were recorded. To standardize the data related to the Ki67 protein, the limit values proposed in the study by Uguen et al. were used in the personal research (<2%, 2–5%, 6–10%, 11–20%, and >20%), given the fact that an elevated Ki67 index was a strong argument for a malignant melanocytic lesion, but a low value did not eliminate the diagnosis of melanoma [32].

This technique for molecular biology testing of the CDKN2A status was implemented at the Center for Research and Development of the Morphological and Genetic Studies of Malignant Pathology (CEDMOG), using dual-color fluorescence in situ hybridization (FISH). The formalin-fixed paraffin-embedded samples of the 23 selected cases were processed with the aid of the chromosome 9 centromere (Table 1) in order to detect the percentage values for p16/CDKN2A deletion.

The initial evaluation was made individually for each of the studied immunohistochemical markers, following the characterization of the tumors included in the present research as CDKN2A-mutated (CDKN2A-mut, lesions with homo- and heterozygous deletions) or CDKN2A-wild type (CDKN2A-wt, cases with monosomy or disomy). 

Based on the data obtained after the application of p16, CD8, and Ki67 antibodies, a comparative analysis of the results drawn from the three individual parameters was elaborated in the groups with familial and multiple primary melanomas, with or without alterations of the CDKN2A gene. With the aid of the correlations made between the values of the immunohistochemical indicators and the data furnished by the specialty literature, scoring systems were considered for each parameter, with different cut-off values established depending on the clinical, diagnostic, or evolutionary significance. The final aim was to design a semi-quantitative algorithm for the evaluation of the CDKN2A mutational context. 

The two sub-groups of patients (synthetically named “CDKN2A-wt” and “CDKN2A-mut”) were afterward randomly divided into two almost equal parts: the first part of each of these categories (the test set) was used to derive the total values of the triple p16-CD8-Ki67 score, while the second part (the validation set) comprised the cases analyzed to confirm the accuracy of the method and the performance of the proposed score. 

For the statistical analysis, Microsoft Excel (Microsoft Office, Redmond, WA, USA) was used, and the *p*-value was considered significant when under 0.05.

## 3. Results

### 3.1. Individual Evaluation of p16, CD8, and Ki67

The individual evaluation of each of the aforementioned immunohistochemical and molecular parameters generated valuable information, as shown in Table 2. 

In the evaluated cohort, three cases of heterozygous deletion, seven patients with homozygous deletion, seven subjects with disomy, and six with CDKN2A monosomy were identified. Thus, the cases with mutated CDKN2A expression totaled 10 patients, while 13 tissue samples revealed the absence of the aforementioned genetic alteration. 

The average value of p16 positivity was 42.30% in the familial and multiple primary melanoma category without CDKN2A mutation and 15.00%, respectively, in the sub-group of patients with homo- or heterozygous deletions of CDKN2A. 

The CD8-positive TILs presented an average value of 5.00% in CDKN2A-mutated tumors, while cases without genetic alterations identified via biomolecular methods were associated with an average percentage of CD8-positive lymphocytes of 19.61%. 

Moreover, the Ki67 expression in subjects with familial and multiple primary melanomas associated with CDKN2A mutations was greater (42.50%) in comparison with CDKN2A-wild type tumors, where the detected average percentage was 35.38% (Table 3). 

Inferior values of p16-positive cells have been identified in the CDKN2A-mutated group, a finding similar to the one extracted after the analysis of CD8-processed samples that determined positivity in a lower percentage of TILs in the category with CDKN2A alterations. Eventually, Ki67 expression was more intense in patients with proven CDKN2A mutation. 

### 3.2. Assessment of Clinical and Genetic Correlations in the Analyzed Cohort

The evaluated cases included 50 primary melanomas, corresponding to 16 patients with multiple primary melanomas and 7 individuals with familial melanoma. Of these, in the category of multiple primary melanoma cases, 43 primary tumors were identified, with a minimum of 2 primary tumors per patient, a maximum of 9 primary tumors per individual, and an average value of 2.68 primary tumors per case.

Within the evaluated cohort, a male:female sex ratio of 1.3:1 was identified. Among CDKN2A-wt cases, eight patients were male (61.53%) and five female (38.46%), while the gender ratio was equal in the group of individuals with CDKN2A deletions.

Regarding the location of the melanomas examined, the anatomical sites were standardized into four main categories: cervical–cephalic extremity; upper extremities; lower extremities; and trunk. Following this standardized nomenclature, the group of CDKN2A-wt patients included a total number of 25 primary tumors, of which 8 were cervical–cephalic (32.00%), 7 located in the upper extremities (28.00%), 2 in the lower extremities (8.00%), and 8 at the truncal level (32.00%). Next, the anatomical distribution of the 25 primary melanomas diagnosed within the analyzed population diagnosed with CDKN2A mutation was evaluated, which presented the following pattern: 11 cervical–cephalic tumors (44.00%); 4 neoplasias in the upper limbs (16.00%); 6 in the lower limbs (24.00%); and 4 cases identified truncally (16.00%). Subsequently, analysis of these data revealed a net prevalence of tumors located on intensely photoexposed areas, such as the cervical–cephalic extremity, among CDKN2A-mutated patients. The diverse location of primary tumors in individuals with CDKN2A mono- or disomy did not show a significant statistical distinction, most frequently being located at the cervical–cephalic and truncal levels.

In addition, the distribution of the primary tumors was also analyzed in relation to the sex of the patients; in this regard, 25 skin melanomas were identified in the male population, distributed as follows: 7 primary cervical–cephalic tumors (28.00%); 6 lesions located on the upper extremities (24.00%); 2 on the lower extremities (8.00%); and 10 at the truncal level (40.00%). The remaining 25 primary tumors corresponding to the female cohort were segregated from the point of view of the anatomical site, as follows: 12 primary neoplasms located on the tegument of the cervical–cephalic extremity (48.00%); 5 primary tumors at the level of the upper extremities (20.00%); 6 in the lower extremities (24.00%); and 2 located in the trunk (8.00%). Following this investigation, it was highlighted that the majority of cases of primary melanoma registered in the male sex at the truncal level and, respectively, at the level of the cervical–cephalic extremity in the female subgroup.

Corroborating the mutational status and anatomic localization correlation data with those of the sex-localization relationship, the logical hypothesis drawn would be the predominance of CDKN2A-wt tumors in male patients and those with CDKN2A deletions in female patients. This assumption was supported by the previously obtained results, which showed a 61.53% percentage of CDKN2A-wt tumors in males and a 50% allocation of CDKN2A-mut melanomas among females.

In the subgroup of 13 patients without CDKN2A mutation, 9 of them were diagnosed with multiple primary melanomas (69.23%), and 4 belonged to families with aggregation of melanoma cases (30.76%). On the other hand, the 10 cases identified as carrying homo- and heterozygous deletions of CDKN2A were segregated into 7 cases of MPM (70%) and 3 individuals with familial melanoma (FM, 30%). Thus, the absence of statistical differences in the MPM-FM distribution was noted, regardless of the mutational signatures associated with the CDKN2A gene.

### 3.3. The Triple p16-CD8-Ki67 Scoring Algorithm for the Distinction between Familial and Multiple Primary Melanomas with and without CDKN2A Mutations

Based on the results obtained via the individual analysis of the parameters considered in the present study, the objective was to create a score whose progressively higher values were superimposed on those cases with an unfavorable prognosis, corresponding to tumors with CDKN2A mutation, as opposed to lower total values applicable to CDKN2A wild-type cases.

In this regard, the percentage of p16-positive cells was stratified by means of a four-class digressive scale as follows: expression present in >50% of cells (score 0); 11–50% (score 1); 1–10% (score 2); and complete absence of p16 immunoreaction (score 3). The significance of the CD8 antibody in the peritumoral lymphocyte population was interpreted according to the evolutionary patterns and responsiveness to treatment noted in the present study group and in previous research; therefore, the cut-off values considered for scoring were intense staining of peritumoral CD8 TILs >60% (score 0), moderate intensity, with 20–60% CD8-positive TILs (score 1), low intensity <20% (score 2), and the absence of peritumoral TILs (score 3). The quantification of the Ki67 proliferation index was based on the use of a progressive-ascending scale with five classes, which determined the neoplastic characterization from less proliferative tumors (score 0) to intensely proliferative ones (score 4): lower than 2% (score 0); 2–5% (score 1); 6–10% (score 2); 11–20% (score 3); and greater than 20% (score 4). In this global context, the total value after performing the proposed score per tumor varied between 0 and 10 (Table 4).

By applying the algorithm described above, the values shown in Table 5 were obtained within the test set.

The triple-score cut-off of 9 was noted to differentiate homozygous CDKN2A deletion melanomas from CDKN2A-wild type and heterozygous deletion tumors.

Analysis of the tumors included in the p16-CD8-Ki67 score validation set revealed the information contained in Table 6.

All eight cases of multiple/familial primary mucocutaneous melanomas with CDKN2A homozygous deletion corresponded to a total score ≥9.

However, false-negative results were identified (three patients with CDKN2A heterozygous deletion, with a total score below 9) and one false-positive case (a CDKN2A-wild type tumor, detected by FISH as monosomy, with scoring values above 9).

Using the appropriate formulas for sensitivity and specificity, these indicators were determined in the present study to establish the accuracy of the method as a tool for predicting homozygous deletions recorded at the level of the CDKN2A gene. The sensitivity of the test represents the rate of true-positive cases, i.e., the proportion of true-positive cases (individuals with homozygous CDKN2A mutation correctly identified by the proposed score) from the total number of patients carrying this gene alteration. Next, the specificity of the algorithm, also referred to as the rate of true-negative cases, signifies the proportion of true-negative cases (patients without CDKN2A homozygous deletion optimally detected by means of the score) from the total number of patients who do not present this genetic condition. In the context where the detection limit of melanomas with homozygous CDKN2A deletion is set to a total value of at least 9, the sensitivity of the test is 100%, and the specificity is 94.11% (Figure 1).

## 4. Discussion

To our knowledge, the present study is the first research of its kind to propose the use of a triple algorithm based on immunohistochemical markers for the distinction between multiple and familial primary melanomas with CDKN2A homozygous deletion versus CDKN2A wild-type and heterozygous deletion cases. Until now, clinical and dermoscopic protocols for nevus–melanoma differentiation have been developed [34,35], as well as a p16-HMB45/MelanA-Ki67 immunohistochemical algorithm created with the same purpose [32], but none of them addressed the prediction of CDKN2A mutational status.

The algorithm composed of three immunohistochemical parameters selected on the basis of their prognostic and evolutionary importance represents a valuable auxiliary diagnostic tool in the detection of multiple and familial primary melanomas with CDKN2A homozygous deletion as a cost-effective measure of mutational status prediction.

Loss of nuclear p16 expression coincided with the presence of tumor ulceration and vascular invasion but not with neoplastic population thickness, histological diameter, or Clark level [36]. The results of a study conducted by Straume et al. indicated that reduced p16 expression might represent an early event in some melanoma cases and was not directly associated with tumor size at diagnosis [37]. In this regard, a strong and independent relationship between absent or minimal p16 immunophenotype and prognosis was noted, indicating a close association with the ability of malignant tumor cells to disseminate from the primary site [32].

The research by Soo et al. published in 2023 reported the identification of a synthetic peptide capable of cell penetration and induction of apoptosis of neoplastic melanoma cells in an efficient and selective manner, with the advantage of lower toxicity to normal melanocytes and absence of human fibroblasts [38]. This peptide is based on the target-binding site of the p16 effector involved in senescence and suppression of melanoma progression, combined with a fraction that facilitates intracellular penetration [39].

The low frequency of CD8-positive TILs constitutes an unfavorable prognostic factor in various types of oncological conditions [40]. In numerous previous studies in the literature, CD8 intensity was quantified as weak (5–20%), moderate (20–60%), and intense (>60%), with consensus in the pathology community [32]. That is why the limits set in the present study in order to design the proposed algorithm were similar to those described above regarding the CD8 indicator.

Melanomas located on the skin at the level of the cervical–cephalic extremity were associated with an increased level of TILs compared to tumors located at the level of the extremities [41], a fact that suggests the involvement of the anatomical site in the progression and aggressiveness associated with this oncological category.

The role of Ki67 as an index of proliferation has been indicated by multiple previous studies, of which a recent meta-analysis reiterated the association of increased expression of this protein with tumor thickness but not with other parameters, such as sex, location, ulceration, or vascular invasion [42]. Additionally, research conducted by Liu et al. indicated the relationship between elevated Ki67 values and lower overall survival (OS) rates in the subgroup of patients diagnosed with melanoma, regardless of geographical region, age, cut-off values of Ki67 marker expression, and duration of follow-up [43].

The association of the three parameters included in the scoring system was carried out in light of the proven utility of p16, CD8, and Ki67 in the evolution of cutaneous and mucosal melanoma cases.

The study by Straume et al. identified a mean proliferation rate assessed by the Ki67 expression of 35% in cases with absent or minimal nuclear p16 expression, in contrast to a mean value of 24% obtained in the subgroup of patients with moderate or intense immunohistochemical p16 staining [36]. Thus, the role of p16 protein integrity in cell cycle inhibition was reiterated, whose changes were also translated by accelerated tumor proliferation, an idea supported by the high percentages of Ki67 [36]. Although the association between the absence of p16 and increased values of Ki67 is proven to be involved in increasing the aggressiveness of melanoma, these indicators also denote an individual role in truncal cutaneous ulcerated melanomas with increased Breslow index and present vascular invasion (in the case of lack of expression of p16) and reduced survival (in the case of Ki67) [36].

The diagnosis of most oncological conditions is performed by pathologists via the microscopic analysis of surgically obtained tissue fragments, and recent scientific explorations have indicated a notable discordance between examiners, reported at values of 25–26% for the distinction between nevi and melanomas [44]. In this sense, avant-garde methods have been implemented, such as deep learning, a technique based on the principle of neural networks, to increase the diagnostic accuracy of lung or breast cancer [45]. The same technique was applied to the histopathological classification of nevi and melanomas, but the rate of variation in diagnostic accuracy was similar to that recorded among human examiners [46].

The increased heterogeneity of immunohistochemical analyses from the point of view of these techniques and quantification methods (manual or automatic, using single or double staining, the number of tumor cell nuclei analyzed, the average value calculated within the entire lesion or exclusively at the level of intensely proliferative hotspots, the quantitative or semi-quantitative approach with various scales) generates challenges in the diagnosis and the standardization in a common histopathological language of the results obtained [47]. Also, the choice of cut-off values for the algorithmic evaluation of multiple and familial primary melanomas has become challenging in light of this disparity of results presented in specialized studies.

Even though CDKN2A heterozygous deletions are sufficient to confer a 67% risk of melanoma development, it was found that the mechanisms responsible for oncogenesis and tumor progression required clarification at the present time [48,49]. Corroborating the data obtained in the personal study, in which a lack of association was noted between the increased values of the triple p16-CD8-Ki67 score and CDKN2A heterozygous losses with the statements related to this genetic alteration extracted from the dermatopathology literature [50], the results derived from the present research can be explained.

In addition, in the review conducted by Levanat et al., it was discovered that a heterozygous CDKN2A gene defect could be detected in a relative of a melanoma patient not yet diagnosed with this condition, which would indicate a clear predisposition to develop melanocytic cutaneous malignancies, but without the obligation for melanoma to occur, however, during their lifetime [51].

Taking into account the higher frequency of heterozygous CDKN2A deletions (36%) in melanoma specimens compared to homozygous losses (16%) [52], the relevance of the triple algorithm for predicting CDKN2A homozygous deletions in patients with MPM and familial melanoma, inapplicable to cases with heterozygous deletion, may be generated precisely by the variation in the incidence and clinical significance that the two different types of genetic alterations carry. In addition, there is emerging evidence regarding the utility of tests to identify CDKN2A deletions in the differential diagnosis of Spitz nevus versus spitzoid melanoma [53]; therefore, this line of research, as well as the applicability of the p16-CD8-Ki67 triple immunohistochemical score, may be successfully explored in the future.

The limitations of this personal study involve the low number of patients who met the inclusion criteria during the evaluated time period. In addition, the cut-off values for each of the three parameters examined immunohistochemically (p16, CD8, Ki67) were established on the basis of previous research aimed at tracing correlations between the immunophenotypic characteristics of cutaneous and mucosal melanoma at diagnosis and the patterns of evolution identified over time; however, the potential for confirmation bias due to the heterogeneity of data currently available in the literature must be acknowledged. The current analysis focused on patients from South-Eastern Romania; therefore, the usefulness of this algorithm must also be tested within populations diagnosed with familial or multiple primary melanomas originating in other geographical areas of Romania and the European continent. Finally, given that different histopathological subtypes of MPM and familial melanoma possess variable behavior via their molecular signatures, the algorithm must be correctly interpreted in relation to CDKN2A homozygous deletion.

As the number of newly diagnosed melanomas in Romania is approximately 1547 new cases annually [54], a fact corroborated by the low percentages of multiple primary melanomas that range between 0.2 and 8.6% [55], as well as those of familial melanomas that follow a similar incidence pattern (1–8%) [56], the sample size included in the present study reflects the incidence and prevalence encountered in the South-Eastern Romania. To standardize and increase the applicability of the p16-CD8-Ki67 scoring system in predicting CDKN2A mutational status in individuals with clinical evidence of multiple and/or familial primary melanomas, studies with a comprehensive design are needed in larger cohorts or its integration into large-scale clinical trials could prove useful.

## 5. Conclusions

In conclusion, p16, CD8, and Ki67 are valuable individual indicators for predicting the evolution of melanoma cases. P16 shows more attenuated or absent expression in the group of patients with CDKN2A mutations, and similarly, the percentage of tumor-infiltrating lymphocytes with positivity for the CD8 immunohistochemical antibody is lower in the group of patients with homo- and heterozygous deletions of CDKN2A, compared to the results obtained in the subgroup of monosomies and disomies. Moreover, the expression of the Ki67 index reported in percentages registered higher values in the context of patients proven by means of the FISH method to be carriers of CDKN2A mutations. Identification of false-negative and false-positive results after the application of the three-tier algorithm suggests the perfectibility of the developed score following the performance of large-scale studies. Corroborating the significance of the three parameters included in the p16-CD8-Ki67 immunohistochemical score, the results derived from this study and the existing data in the literature, the evaluation of the utility of the above-mentioned algorithm in the atypical nevus–melanoma distinction constitutes a valid, future research direction.

## Figures and Tables

**Figure 1 diagnostics-14-00813-f001:**
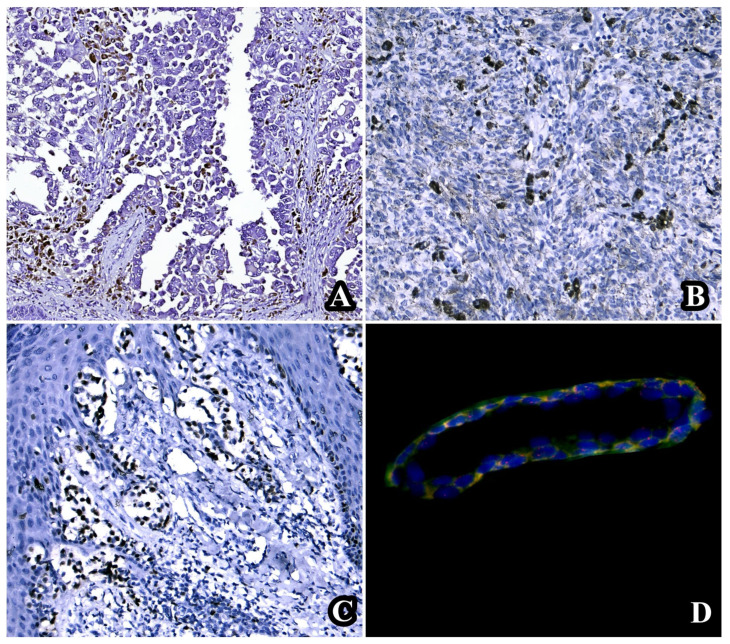
Iconographic representation of a classic case of melanoma with homozygous deletion and the maximum score obtained following the application of the triple algorithm. (**A**) Absence of p16 immunohistochemical expression (p16 × 20). (**B**) Immunohistochemical absence of CD8+ TILs (CD × 20). (**C**) Ki67 index value of 40% (Ki67 × 20). (**D**) CDKN2A homozygous deletion revealed by FISH technique.

**Table 1 diagnostics-14-00813-t001:** The panel of immunohistochemical antibodies and genetic markers used in this study.

Marker	Clone	Manufacturer	Dilution	Host, Clonality/Additional Materials
P16	MX007	Master Diagnostica	Ready-to-use (RTU) 7 mL	Mouse, monoclonal
CD8	SP16	Master Diagnostica	RTU 7 mL	Rabbit, monoclonal
Ki67	SP6	Biocare	RTU 6 mL	Rabbit, monoclonal
SPEC CDKN2A/CEN 9 Dual Color Probe	ZytoLight^®^	ZytoVision GmbH, Bremerhaven, Germany	RTU 0.2 mL	ZytoLight FISH Implementation Kit

**Table 2 diagnostics-14-00813-t002:** Characterization of the analyzed patients from the point of view of the immunohistochemical (p16, CD8, Ki67) and molecular (CDKN2A mutational status) markers.

Patient Code	P16	CD8	Ki67	CDKN2A Analysis
SV001	50%	0%	40%	Disomy
MS002	50%	20%	20%	Heterozygous deletion
IM003	50%	0%	25%	Heterozygous deletion
HV004	0%	0%	40%	Homozygous deletion
CV005	50%	20%	60%	Heterozygous deletion
RI006	0%	5%	60%	Homozygous deletion
PC007	50%	0%	10%	Disomy
BN008	50%	20%	10%	Monosomy
MR009	50%	70%	60%	Monosomy
MI010	0%	5%	60%	Homozygous deletion
PS011	50%	10%	10%	Disomy
AM012	50%	40%	70%	Disomy
GN013	0%	0%	40%	Homozygous deletion
CM014	0%	0%	50%	Homozygous deletion
CN015	50%	5%	50%	Disomy
GU016	50%	10%	10%	Disomy
SM017	50%	0%	60%	Monosomy
MS018	0%	0%	20%	Homozygous deletion
VS019	0%	0%	80%	Monosomy
LV020	0%	90%	20%	Monosomy
PD021	50%	5%	30%	Disomy
PG022	50%	5%	10%	Monosomy
TV023	0%	0%	50%	Homozygous deletion

**Table 3 diagnostics-14-00813-t003:** Summary of the obtained results.

Marker	CDKN2A-Wild Type Melanomas (Average Value)	CDKN2A-Mutated Melanomas (Average Value)	*p*-Value
P16	42.30%	15.00%	0.009045612
CD8	19.61%	5.00%	0.118534718
Ki67	35.38%	42.50%	0.430718946

**Table 4 diagnostics-14-00813-t004:** The proposed triple-parameter scoring system.

Parameter	Score 0	Score 1	Score 2	Score 3	Score 4	Total Score
P16 (positive cells)	>50%	11–50%	1–10%	0%	-	0–10
CD8+ peritumoral TILs	>60%	20–60%	<20%	0%	-	
Ki67 (proliferative index)	<2%	2–5%	6–10%	11–20%	>20%	

**Table 5 diagnostics-14-00813-t005:** Values of the triple-parameter algorithm obtained in the test set.

Patient Code	CDKN2A Analysis	Total Value of the Triple p16-CD8-Ki67 Score
SV001	Disomy	8
PC007	Disomy	6
BN008	Monosomy	4
MR009	Monosomy	5
PS011	Disomy	5
SM017	Monosomy	8
MS002	Heterozygous deletion	5
IM003	Heterozygous deletion	8
HV004	Homozygous deletion	10
CV005	Heterozygous deletion	5
RI006	Homozygous deletion	9

**Table 6 diagnostics-14-00813-t006:** Values of the triple-parameter algorithm obtained in the validation set.

Patient Code	CDKN2A Analysis	Total Value of the Triple p16-CD8-Ki67 Score
AM012	Disomy	6
CN015	Disomy	7
GU016	Disomy	5
VS019	Monosomy	10
LV020	Monosomy	6
PD021	Disomy	7
PG022	Monosomy	5
MI010	Homozygous deletion	9
GN013	Homozygous deletion	10
CM014	Homozygous deletion	10
MS018	Homozygous deletion	9
TV023	Homozygous deletion	10

## Data Availability

The data generated in the present study are included in the figures and/or tables of this article.

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
