# Peer review of "P16-CD8-Ki67 Triple Algorithm for Prediction of CDKN2A Mutations in Patients with Multiple Primary and Familial Melanoma"

_diagnostics, 2024, doi:10.3390/diagnostics14080813_

Round 1

Reviewer 1 Report

Comments and Suggestions for Authors

This interesting article proposes an innovative algorithm that could have useful implications in clinical practice in the future. Although the sample size is small, future studies on the topic will be able to test the proposed algorithm.

Comments on the Quality of English Language

English language is fine, I didn’t detect any major flaws 

Reviewer 2 Report

Comments and Suggestions for Authors

Manuscript entitled"P16-CD8-Ki67 Triple Algorithm for Prediction of CDKN2A Mutations in Patients with Multiple Primary and Familial Melanoma"

1. IHC is semi-quantitative and is of large variety between labs. It is not an ideal way to perform three IHC as a screen tool.

2. The authors should include more cases, with sporadic or familiar cases.

3. The work can only be significance if the authors can test a wider cancer spectrum.

Comments on the Quality of English Language

Acceptable

Reviewer 3 Report

Comments and Suggestions for Authors

The manuscript by Nurla et al., entitled “ P16-CD8-Ki67 Triple Algorithm for Prediction of CDKN2A Mutations in Patients with Multiple Primary and Familial Melanoma “  is designed to evaluate the reliability of  P16-CD8-Ki67 p16, CD8, and Ki67 individually serve as a predictive marker for  multiple and  familial primary melanomas with CDKN2A homozygous deletion. The manuscript is well written and the data are well presented Accordingly the manuscript can be published in the present Form.

Comments on the Quality of English Language

The english langauge is acceptable

Round 2

Reviewer 2 Report

Comments and Suggestions for Authors

Apologize but there is no improvement made

Comments on the Quality of English Language

Acceptable